# Prognostic Value of Plasma Presepsin and Pneumonia Severity Index in Patients with Community-Acquired Pneumonia in the Emergency Department

**DOI:** 10.3390/medicina58111504

**Published:** 2022-10-22

**Authors:** Kyeong-Ryong Lee, Dae-Young Hong, Jin-Hui Paik, Hyun-Min Jung

**Affiliations:** 1Department of Emergency Medicine, Konkuk University School of Medicine, Seoul 05030, Korea; 2Department of Emergency Medicine, Inha University School of Medicine, Incheon 22332, Korea

**Keywords:** presepsin, pneumonia, biomarkers, disease severity, mortality

## Abstract

*Background and Objectives*: Presepsin (PSS) is an independent predictor for estimating disease severity and prognosis in septic patients. Few studies have reported the associations between plasma PSS and the severity and prognosis in patients with community-acquired pneumonia (CAP). We investigated whether a high plasma PSS level was associated with 30-day mortality in CAP patients. *Materials and Methods*: This retrospective single-center study was conducted in an emergency department. The PSS level was measured in 211 adult CAP patients admitted to the hospital and followed for up to 30 days. We recorded the pneumonia severity index (PSI) and the CURB-65 score. The primary outcome was death from any cause within 30 days. *Results*: The plasma PSS levels were significantly elevated in the high-risk group (PSI > 130) compared with the low- (PSI < 91) or moderate-risk groups (PSI 91–130). Forty-four patients (20.9%) died within 30 days of admission. Non-survivors had significantly higher plasma PSS levels than survivors among CAP patients: 1083 (697–1736) pg/mL vs. 385 (245–554) pg/mL (*p* < 0.001). The area under the curve (AUC) to predict 30-day mortality was highest for PSS (0.867), followed by procalcitonin (0.728) and lactate (0.616). The cutoff level of plasma PSS for 30-day mortality was >754 pg/mL. The combination of PSI and plasma PSS level improved the predictive ability for 30-day mortality (AUC = 0.892). Cox regression analysis showed that higher PSS levels (>754 pg/mL) and higher PSI (>126) were associated with 30-day mortality in CAP patients (hazard ratios of 19.472 and 6.375, respectively). *Conclusion*: Elevated plasma PSS is associated with severity and 30-day mortality in hospitalized CAP patients. Combining plasma PSS level and PSI could significantly improve the predictive ability of PSS for 30-day mortality.

## 1. Introduction

Community-acquired pneumonia (CAP) is a common lower respiratory infection in an emergency department (ED) with high morbidity worldwide. Despite improvements in early diagnosis and new antimicrobial agents, CAP still causes serious complications and hospital deaths [1]. Timely and accurate severity assessment and prognosis prediction are recommended for the proper and specific management of patients with CAP.

The CURB-65 (confusion, urea, respiratory rate, blood pressure, and age ≥ 65 years) score and the pneumonia severity index (PSI) have been commonly used to assess the disease severity and outcomes in patients with CAP. Compared with the PSI, which requires the evaluation of 20 parameters, the CURB-65 score is relatively simple and more convenient to use in a crowded ED. However, the CURB-65 score does not directly reflect the presence or absence of comorbidities and may underestimate the mortality risk in elderly patients.

In addition to scoring systems, various biomarkers, such as complete blood counts and levels of interleukin-6, C-reactive protein (CRP), and procalcitonin, have been studied for early diagnosis, severity assessment, and outcome prediction in patients with CAP [2,3,4,5,6].

The cluster of differentiation 14 (CD14) is present on the surface of macrophages, monocytes, and granulocytes. Presepsin (PSS), a newly emerging biomarker for sepsis, is generated from soluble CD14 by the proteolytic action of circulating cathepsin D [7]. Studies have demonstrated that PSS could be reliable for estimating severity and predicting mortality in septic patients [8,9]. However, few studies have focused on PSS as a biomarker for predicting mortality or suggested a cutoff level for predicting mortality in CAP patients [10,11]. Therefore, this study aimed to determine a cutoff level of plasma PSS as a prognostic biomarker for 30-day mortality and further investigate the predictive ability of the combination of plasma PSS level and PSI in patients with CAP.

## 2. Materials and Methods

### 2.1. Study Design

This retrospective single-center study was conducted in the ED of Konkuk University Medical Center in Seoul, Korea, from December 2020 to May 2022. Included were consecutive adult CAP patients (≥19 years) admitted to the hospital via the ED. The exclusion criteria were: patients with hospital-acquired or healthcare-associated pneumonia or coronavirus disease 2019, end-stage kidney disease, or malignancy; those transferred from another hospital; and those admitted for palliative treatment.

According to the Infectious Disease Society of America and American Thoracic Society consensus guidelines, patients with CAP and respiratory symptoms and signs were diagnosed with pulmonary infiltration using chest radiography [12].

The study was approved by the Ethics Committee of the Konkuk University Medical Center. The informed consent requirement was waived due to the study’s retrospective nature.

### 2.2. Variables

Venous blood samples were immediately collected after arrival at the ED. The plasma PSS levels were determined using a chemiluminescent immunoassay (PATHFAST^TM^, LSI Medience, Tokyo, Japan).

CURB-65 score and PSI were calculated using the recorded information in the ED: age, sex, comorbidities, vital signs, physical examination, results of laboratory examinations, and radiological findings. The enrolled patients with CAP were divided into high-risk (PSI > 130), moderate-risk (PSI 91–130), and low-risk (PSI < 91) groups.

The primary outcome of this study was death by any cause within 30 days after admission.

### 2.3. Statistics

All medical records and information of patients were anonymized before the analysis.

Categorical variables were presented as frequency and percentages, and chi-square tests were performed for analysis. Continuous variables were presented as mean and standard deviation or median and 25th–75th percentiles. The Mann–Whitney U test or Student’s *t*-test were used to compare survivors and non-survivors, and the Kruskal–Wallis test or one-way analysis of variance were used for multiple-group comparison (low-, moderate-, and high-risk groups). The Spearman correlation test was conducted to assess the relationship between plasma PSS and other continuous variables.

To compare the predictive value of the variables for 30-day mortality, receiver operating characteristic (ROC) curves were recorded, and the area under the curve (AUC) was calculated. Cox regression analyses were used to evaluate the significance of prognostic factors for 30-day mortality. Statistical analyses were performed using R-4.1.1 for Windows (http://www.r-project.org), SPSS Statistics (version 27; IBM Corp., Armonk, NY, USA) and MedCalc (version 20.011; MedCalc Software, Ostend, Belgium). Two-sided *p*-values < 0.05 were considered statistically significant.

## 3. Results

### 3.1. Demographics and Clinical Information

From December 2020 to May 2022, 305 CAP patients were admitted via the ED. Of those patients, 92 were excluded due to the exclusion criteria, and 2 were excluded because they were lost during the 30-day follow-up period. Finally, 211 patients were included in this study (Figure 1).

The median age was 78 (66–84) years, and 123 (58.3%) patients were male. The mean PSI at ED admission was 109 ± 21. All enrolled patients had a 30-day hospital mortality rate of 20.9% (44/211). The demographic characteristics, comorbidities, and clinical presentations are summarized in Table 1.

Spearman analysis of the correlation between plasma PSS level and age, PSI, CURB-65 score, and levels of hsCRP, lactate, and procalcitonin revealed the positive correlation coefficients of 0.164, 0.302, 0.339, 0.212, 0.227, and 0.375, respectively (all *p* < 0.05). However, there was no correlation between plasma PSS level and white blood cell (WBC) count (*p* = 0.952).

### 3.2. Association of PSS Levels and Disease Severity

All CAP patients were divided into three risk groups according to their PSI at the time of ED arrival. The high-risk group (PSI > 130) accounted for 28.9% (61/211) of all patients with CAP. The mean and median laboratory results for the risk groups are presented in Table 2.

The lactate, procalcitonin, and PSS levels in the high-risk group at ED admission were significantly higher than in the moderate- and low-risk groups. However, according to the PSI risk groups, the hsCRP levels and WBC counts did not gradually become higher with increasing severity.

### 3.3. Association of PSS Levels and Clinical Outcomes

The initial mean WBC count and median hsCRP, lactate, procalcitonin, and PSS levels in survivors and non-survivors are shown in Table 3. The median PSS level was significantly higher in the non-survivors than in the survivors (Figure 2). The median procalcitonin and lactate levels in the non-survivors were also significantly higher than in the survivors (*p* < 0.001 and *p* = 0.020, respectively). However, the non-survivors’ hsCRP level and WBC counts were not significantly different from those in the survivors.

### 3.4. Predictive Mortality Value of PSS Levels

The AUC for the plasma PSS level, CURB-65 score, PSI, WBC count, hsCRP, lactate, and procalcitonin for predicting 30-day hospital mortality are presented in Table 4.

The AUC of plasma PSS level was 0.867 (95% CI: 0.814–0.910, *p* < 0.001), which was higher than those of lactate and procalcitonin (Figure 3). Therefore, the optimal cutoff value of the plasma PSS level was >754 pg/mL (sensitivity of 72.7% and specificity of 91.6%) for predicting 30-day mortality in CAP patients.

We evaluated the incremental benefit of combining the plasma PSS levels with the scoring systems. The combination of plasma PSS and PSI improved the predictive ability for 30-day mortality in CAP patients (AUC = 0.892, Youden’s index 0.664, *p* = 0.031). However, the plasma PSS level and CURB-65 score did not significantly improve the predictive value, with Youden’s index increasing from 0.643 to 0.648 (AUC = 0.868, *p* = 0.468).

Age, sex, comorbidities (diabetes mellitus, hypertension, cerebrovascular disease, chronic liver disease, congestive heart failure, chronic lung disease, and chronic renal disease), CURB-65 score > 2, PSI, bacteremia, intensive care unit (ICU) admission, WBC count, and levels of hsCRP, lactate, procalcitonin, and PSS were included in a Cox regression analysis (Table 5). In the univariate analysis, age, CURB-65 score > 2, PSI > 126, ICU admission, hsCRP level > 8.63 mg/dL, lactate level > 2.81 mmol/L, procalcitonin level > 0.22 ng/mL, and PSS level > 754 pg/mL were significantly associated with 30-day mortality. Multivariate analysis showed that the plasma PSS level > 754 pg/mL (HR 19.472, 95% CI: 7.262–52.209) and PSI > 126 (HR 6.375, 95% CI: 2.374–17.122) alone were independent predictors of 30-day mortality in CAP patients.

## 4. Discussion

In this study, we investigated the relationship between the initial plasma PSS level at ED admission and the ability to predict 30-day mortality in patients with CAP. In our study, the initial plasma PSS level was significantly elevated in the high-risk group (PSI > 130) and non-survivors. As a result, the plasma PSS level was a useful predictor for 30-day mortality in CAP patients, and its cutoff value was >754 pg/mL. In addition, the predictive ability of plasma PSS level improved when combined with PSI.

PSS is a relatively novel inflammatory biomarker; its levels increase earlier than other biomarkers in systemic infections [13]. Procalcitonin levels begin to increase within 2 to 4 h after infection. Plasma PSS levels increase earlier and more rapidly than procalcitonin after infection [14,15]. Most studies on PSS were mainly focused on patients with sepsis [8,9,16,17], and few studies focused on patients with CAP. A previous study reported plasma PSS level as a reliable biomarker for the initial discrimination between bacterial CAP and active pulmonary tuberculosis. The combination of the plasma PSS level with the CURB-65 score could significantly improve discriminative accuracy [18]. In the study by Halıcı, the serum PSS levels in acute exacerbation of chronic obstructive pulmonary disease (COPD) patients with pneumonia were significantly higher than in patients with only acute exacerbation of COPD [19]. Moreover, tracheal aspirate PSS was a useful early diagnostic marker of pneumonia in intubated newborns [20].

It has been demonstrated that plasma PSS is related to disease severity in patients with CAP. Liu et al. studied 573 patients with CAP and found that the median plasma PSS level was significantly elevated in patients with a high CURB-65 score (≥3) than in those with a low CURB-65 score (0–2) [10]. Ham and Song studied 90 patients with CAP and reported that the median PSS level was significantly higher in patients with a high PSI (≥71) than in those with a low PSI (<71) (655.0 ng/L vs. 382.5 ng/L, *p* = 0.001) [21]. In accordance with previous studies, the plasma PSS level at ED admission was significantly elevated in the high-risk patients (PSI > 130) compared with low- and moderate-risk patients (PSI 91–130 and PSI < 91, respectively), and it was associated with 30-day mortality in our study. In addition, the present study showed that procalcitonin and lactate levels at ED admission, but not the WBC count and hsCRP levels, were associated with the severity of CAP.

We investigated the present study’s relationship between plasma PSS levels with scoring systems and inflammatory biomarkers. The plasma PSS level correlated positively with the CURB-65 score, PSI, and levels of procalcitonin, lactate, and hsCRP. However, plasma PSS level had no association with WBC count. Similar to our findings, Ugajin et al. showed that the plasma PSS level was positively correlated with CRP (r = 0.375) and PSI (r = 0.384) [11].

We compared the plasma PSS levels in survivors and non-survivors over a 30-day follow-up period. The 30-day mortality rate was 20.9% (44/211). Among the 44 non-survivors, 70.5% (31/44) of the patients were in the high-risk group (PSI > 130) and only four patients were in the low-risk group (PSI < 91). The plasma PSS, procalcitonin, and lactate levels were significantly higher in non-survivors than in survivors, but hsCRP levels and WBC count were not significantly different. Similarly, previous studies reported that deceased patients had higher PSS levels at ED arrival than survivors [10,11]. Klouche et al. studied 58 ICU-admitted patients with severe CAP. They found that the median PSS level was 1734 (1014–3128) pg/mL in non-survivors, which was significantly higher than in survivors (871 (449–1828) pg/mL) [22]. These values are much higher than those reported in other studies. This can be explained by the differences in the study protocol and inclusion criteria, especially the inclusion of ICU-admitted patients with severe CAP.

Plasma PSS levels tend to increase with age [16]. In our study, the plasma PSS levels had a significant but weakly positive correlation with age (r = 0.164). In the univariate analysis, age was an independent predictor of 30-day mortality. However, age was not a predictive variable; only the plasma PSS level and the PSI were independent predictors in the multivariate analysis. Liu et al. also demonstrated that the plasma PSS level and the CURB-65 score were independent predictors of 28-day mortality in CAP [10]. Therefore, the PSI and CURB-65 scores were included as variables in our present study, but Liu et al. did not include PSI as a variable.

Regarding the ROC in the prediction of 30-day mortality, the AUC, sensitivity, and specificity were 0.867, 72.7%, and 91.6%, respectively, when the optimal plasma PSS cutoff value was set to > 754 pg/mL. At the cutoff value of each variable, procalcitonin was the most sensitive (83.3%) and PSS was the most specific (91.6%). The AUC values of the CURB-65 score, PSI, and procalcitonin were 0.717, 0.813, and 0.728, respectively. Plasma PSS level had an excellent predictive ability, and the AUC value of PSS was higher than those of CURB-65 score, PSI, and procalcitonin. The PSI showed equal sensitivity and lower specificity than the PSS level for the 30-day mortality prediction. The AUC value of the plasma PSS level in the present study was higher than in the studies by Liu et al. (AUC for 28-day mortality = 0.672) and Ugajin et al. (AUC for 30-day mortality = 0.742) [10,11]. Moreover, our findings showed that the combination of the plasma PSS level and PSI significantly improved the predictive ability, and in this case, the AUC value was 0.892. This AUC value was higher than the AUC of the plasma PSS level alone or PSI alone. In conjunction with the PSS level, Liu et al. also found that the CURB-65 score significantly increased the AUC value for 28-day mortality [10]. In the present study, there was no significant difference in the predictive ability between PSS alone and PSS level combined with the CURB-65 score.

The present study has some limitations. First, it was conducted in only a single urban ED. Only a limited number of hospitalized patients with CAP were included, and 76% were elderly patients over 65. Therefore, the findings cannot be generalized and applied to all CAP patients. Second, plasma PSS levels were measured only once based on the initial values obtained at the time of ED arrival and not during the follow-up after hospital admission. Third, measuring the plasma PSS level is not cheap; thus, it is difficult to implement this test in all clinical settings.

## 5. Conclusions

We found that plasma PSS levels at ED arrival were significantly elevated in patients with a high PSI and in non-survivors. Moreover, the plasma PSS level was a reliable marker for assessing severity and may be a potential predictor of 30-day mortality in adult patients with CAP. Combining the plasma PSS level with PSI could significantly improve the predictive ability of plasma PSS level, which may help with making clinical judgments in the management of patients with CAP.

## Figures and Tables

**Figure 1 medicina-58-01504-f001:**
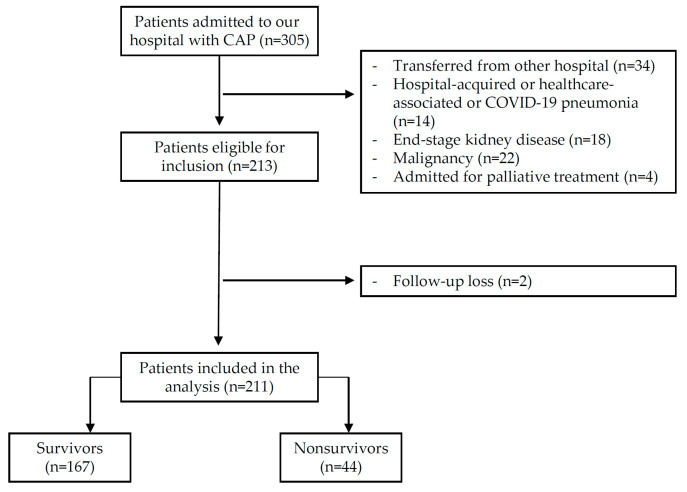
The study flow chart.

**Figure 2 medicina-58-01504-f002:**
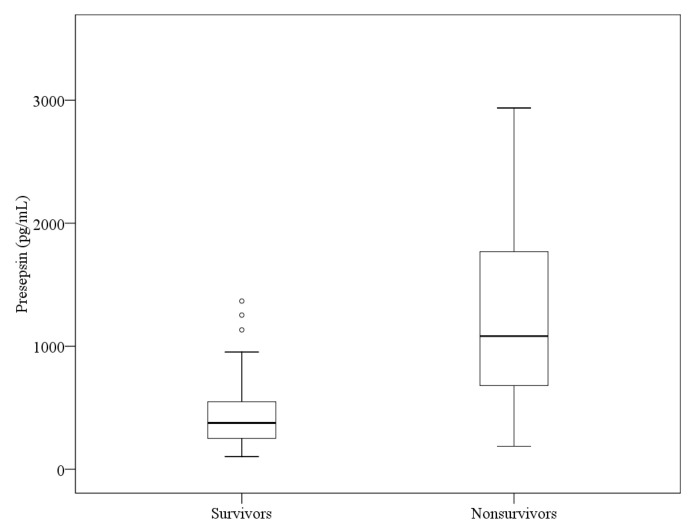
Difference of presepsin levels between survivors and non-survivors.

**Figure 3 medicina-58-01504-f003:**
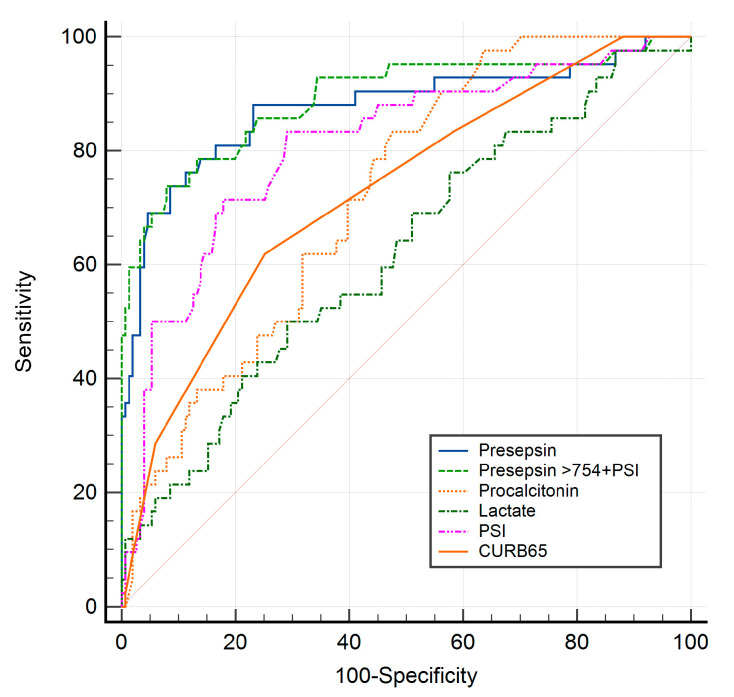
The ROC curves of presepsin, procalcitonin, lactate, PSI, CURB-65 score, and combination of presepsin > 754 pg/mL and PSI for 30-day mortality.

**Table 1 medicina-58-01504-t001:** Demographics and clinical characteristics of the enrolled CAP patients.

Variable	All (n = 211)	Outcome		*p*-Value
		Survivors (n = 167)	Non-Survivors (n = 44)	
Age, y	78 (66–84)	77 (64–84)	80 (73–85)	0.067
Male, n (%)	123 (58.3%)	98 (58.7%)	25 (56.8%)	0.477
Comorbidities, n (%)				
Diabetes mellitus	65 (30.8%)	51 (30.5%)	14 (31.8%)	0.502
Hypertension	96 (45.5%)	79 (47.3%)	17 (38.6%)	0.196
Cerebrovascular disease	77 (36.5%)	60 (35.9%)	17 (38.6%)	0.435
Chronic liver disease	10 (4.7%)	6 (3.6%)	4 (9.1%)	0.131
Chronic lung disease	14 (6.6%)	10 (6.0%)	4 (9.1%)	0.329
Congestive heart disease	8 (3.8%)	7 (2.4%)	1 (2.3%)	0.475
Clinical presentation				
Systolic blood pressure	125 (105–142)	127 (107–143)	116 (88–144)	0.067
Diastolic blood pressure	70 (59–80)	71 (62–81)	66 (53–78)	0.020
Pulse rate	103 ± 21	102 ± 21	106 ± 22	0.275
Respiratory rate	20 (19–24)	20 (20–23)	23 (20–30)	0.012
Body temperature	37.5 ± 1.1	37.6 ± 1.1	37.1 ± 1.1	0.027
CURB-65 score	2 (1–3)	2 (1–3)	3 (2–4)	<0.001
Pneumonia severity index	109 ± 37	100 ± 33	141 ± 33	<0.001
Bacteremia, n (%)	47 (22.3%)	35 (21.0%)	12 (27.3%)	0.241
ICU admission, n (%)	69 (32.7%)	40 (24.0%)	29 (65.9%)	<0.001

**Table 2 medicina-58-01504-t002:** Differences of laboratory results according to the risk groups.

Variable	Low Risk (PSI < 91)	Moderate Risk (PSI 91–130)	High Risk (PSI > 130)
Number, n	71	79	61
Mortality, n (%)	4 (5.6%) ^b,c^	9 (11.4%) ^a,c^	31 (50.8%) ^a,b^
WBC count, ×10^3^/μL	11.8 ± 4.8	11.3 ± 5.0	12.2 ± 6.7
hsCRP level, mg/dL	8.37 (2.74–14.93)	9.41 (4.94–21.08)	10.14 (4.50–19.80)
Lactate level, mmol/L	1.81 (1.21–2.48) ^b,c^	2.23 (1.56–3.24) ^a,c^	3.07 (2.09–5.16) ^a,b^
Procalcitonin level, ng/mL	0.14 (0.05–0.30) ^b,c^	0.26 (0.10–1.03) ^a,c^	0.78 (0.30–5.58) ^a,b^
Presepsin level, pg/mL	295 (214–488) ^b,c^	422 (298–583) ^a,c^	720 (525–1318) ^a,b^

^a^*p* < 0.05, vs. PSI < 91; ^b^
*p* < 0.05, vs. PSI 91–130; ^c^
*p* < 0.05, vs. PSI > 130.

**Table 3 medicina-58-01504-t003:** Differences in laboratory results by survival status.

Variable	Survivors (n = 167)	Non-Survivors (n = 44)	*p*-Value
WBC count, ×10^3^/μL	11.6 ± 5.0	12.3 ± 7.0	0.471
hsCRP level, mg/dL	8.36 (3.30–17.68)	12.45 (4.89–20.68)	0.063
Lactate level, mmol/L	2.22 (1.51–3.13)	2.77 (1.89–4.23)	0.020
Procalcitonin level, ng/mL	0.22 (0.06–0.79)	0.70 (0.27–12.76)	<0.001
Presepsin level, pg/mL	385 (245–554)	1083 (697–1736)	<0.001

**Table 4 medicina-58-01504-t004:** AUCs for predicting 30-day mortality.

Variable	AUC (95% CI)	Cutoff Value	Sensitivity (%)	Specificity (%)	Youden’s Index	*p*-Value
CURB-65 score	0.717 (0.652–0.777)	>2	61.4	73.7	0.350	<0.001
PSI	0.813 (0.754–0.863)	>126	72.7	82.0	0.548	<0.001
WBC count	0.511 (0.442–0.581)	>6.3 × 10^3^/μL	75.0	15.0	0.100	0.831
hsCRP	0.591 (0.522–0.658)	>8.63 mg/dL	71.7	53.3	0.260	0.050
Lactate	0.616 (0.544–0.683)	>2.81 mmol/L	48.8	72.2	0.210	0.020
Procalcitonin	0.728 (0.661–0.789)	>0.22 ng/mL	83.3	52.8	0.362	<0.001
Presepsin	0.867 (0.814–0.910)	>754 pg/mL	72.7	91.6	0.643	<0.001

**Table 5 medicina-58-01504-t005:** Variables associated with 30-day mortality were identified using Cox regression analysis.

Variable	Univariate Analysis		Multivariate Analysis	
	HR (95% CI)	*p*-Value	HR (95% CI)	*p*-Value
Age	1.029 (1.004–1.054)	0.021		
Sex	1.071 (0.590–1.944)	0.822		
Comorbidity	1.596 (0.712–3.581)	0.257		
CURB-65 score > 2	4.580 (2.277–9.214)	<0.001		
PSI > 126	12.178 (5.626–26.358)	<0.001	6.375 (2.374–17.122)	<0.001
Bacteremia	1.429 (0.736–2.774)	0.306		
ICU admission	4.909 (2.628–9167)	<0.001		
WBC count > 6.3 × 10^3^/μL	0.544 (0.275–1.077)	0.081		
hsCRP level > 8.63 mg/dL	2.700 (1.390–5.242)	0.003		
Lactate level > 2.81 mmol/L	2.363 (1.299–4.300)	0.005		
Procalcitonin level > 0.22 ng/mL	4.798 (2.130–10.808)	<0.001		
Presepsin level > 754 pg/mL	15.336 (7.842–29.991)	<0.001	19.472 (7.262–52.209)	<0.001

## Data Availability

The data that support the findings of this study are available on request to the corresponding author.

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
