# Peer review of "Prognostic Value of Plasma Presepsin and Pneumonia Severity Index in Patients with Community-Acquired Pneumonia in the Emergency Department"

_medicina, 2022, doi:10.3390/medicina58111504_

Round 1

Reviewer 1 Report

Thank you for the opportunity to review this well conducted and interest study about the prognostic properties of presepsin in CAP-patients in the ED.

General comment: 

The study describes the prognostic value of a biomarker, which is fine – but not very original and perhaps not very clinically relevant. More studies should investigate the diagnostic value of biomarkers, e.g. the value of diagnosing CAP in dyspnoeic patients.

However, the authors argue satisfactory for the importance of a prognostic biomarkers in their population.

Materials and Methods

Consent was waved by the review board, even though blood samples were drawn from the participants, and presepsin analysed. The authors should describe the ethical evaluation, and how this is in accordance with the declaration of Helsinki.

The blood samples were “immediately obtained at the time of ED arrival”. However, the inclusion criterion requires a new infiltrate on chest x-ray. Were the blood samples drawn before the diagnosis? In that case there must be several patients with blood samples drawn not included in this study? If the blood samples were drawn after diagnoses, I suggesting rephrasing this paragraph.

A negative control group (if available) would strengthen the study

The article should include a flowchart showing how many patients were screened, how many fulfilled the inclusion criterions, and how many were included in the final analysis.

Paragraph 3.4

(Line 184) “The combination of plasma PSS level and CURB-65 score did not significantly improve the predictive value, with the Youden’s index only 186 increasing from 0.643 to 0.648 (AUC = 0.868, 95% CI 0.815–0.910, P < 0.001).” Did not significantly improve – but the P-values is <0.001. Is this correct?

Author Response

1. Consent was waved by the review board, even though blood samples were drawn from the participants, and presepsin analysed. The authors should describe the ethical evaluation, and how this is in accordance with the declaration of Helsinki.
 Reply: There was a typo in the manuscript. This study was conducted in accordance with the Declaration of Helsinki and ethical requirements. Written informed consent was waived owing to the retrospective characteristics of this study. Corrected the misspelled word ‘waved’ to ‘waived’ in the revised manuscript.

2. The blood samples were “immediately obtained at the time of ED arrival”. However, the inclusion criterion requires a new infiltrate on chest x-ray. Were the blood samples drawn before the diagnosis? In that case there must be several patients with blood samples drawn not included in this study? If the blood samples were drawn after diagnoses, I suggesting rephrasing this paragraph. A negative control group (if available) would strengthen the study.
Reply: Thank you for your comment. In Republic of Korea, national health insurance is implemented, so anyone can receive various medical benefits at a relatively low cost compared to other countries. For this reason, it is possible to test various inflammatory biomarkers from the early stage of diagnosis even in the emergency department. Presepsin concentration measurement is also performed in a variety of patients, and the diagnosis is often made after the presepsin concentration has been obtatined. This study was conducted as a retrospective analysis of patients with community acquired pneumonia among patients with various disease for which presepsin concentrations were obtained in the emergency department. Blood samples were obtained immediately upon arrival at the emergency department for all patients included in the study. We ask for your kind understanding.

3. The article should include a flowchart showing how many patients were screened, how many fulfilled the inclusion criterions, and how many were included in the final analysis.
 Reply: We have added flow chart (Figure 1. The study flow chart).

4. (Line 184) “The combination of plasma PSS level and CURB-65 score did not significantly improve the predictive value, with the Youden’s index only 186 increasing from 0.643 to 0.648 (AUC = 0.868, 95% CI 0.815–0.910, P < 0.001).” Did not significantly improve – but the P-values is <0.001. Is this correct?
Reply: The P value was corrected to the comparative analysis of PSS vs PSS + PSI, and PSS vs PSS + CURB65, respectively. In addition, the revised manuscript was corrected as there was a difference in the AUC value of PSS and PSI (Line 272).

Reviewer 2 Report

The authors comprehensively compared the value of different biomarkers in predicting the illness severity and 30-day mortality of CAP patients in the emergency department through a single-center study. The results showed elevated plasma presepsin levels are associated with disease severity and 30-day mortality in hospitalized patients with CAP. Combining plasma presepsin level and psi could significantly improve the predictive ability of plasma presepsin for 30-day mortality. It is an interesting study, but I have the following questions.

1、 This is a retrospective study. However, it is mentioned in the methodology that the specimen was pre-stored at -70 °C, which indicates a prospective study. They are contradictory. The author needs to confirm the type of the study and whether written informed consent was obtained from patients before obtaining the plasma samples.

2、 In this study, 92 patients (30%) were not included in the analysis. Please add a description of the reasons in detail for exclusion.

3、 It is recommended to add PSI, CURB-65, the combination of presepsin level >754 pg/mL and PSI, and the combination of presepsin level >754 pg/mL and PSI for prediction of 30-day mortality in Figure 2.

4、 It is recommended that the methodology used to compare laboratory results between low, moderate, and high risk be described in Table 2.

5、 In Table 5, the underlying disease was included in the analysis as a risk factor, please specify which underlying conditions were included in the analysis.

6、 In Table 4 and Table 5, the cut-off value of PSI is different, why?

Author Response

1. This is a retrospective study. However, it is mentioned in the methodology that the specimen was pre-stored at -70 °C, which indicates a prospective study. They are contradictory. The author needs to confirm the type of the study and whether written informed consent was obtained from patients before obtaining the plasma samples.
Reply: Thank you very much for your comment. This study was conducted retrospectively. The review board considered informed consent unnecessary owing to the characteristics of retrospective analysis. In addition, there was some misunderstanding in the process of inquiring the laboratory medicine staff about the method of measuring the presepsin concentration. Blood samples obtained in the emergency department were immediately sent to a clinical laboratory (blood test room) for concentration measurements.

2. In this study, 92 patients (30%) were not included in the analysis. Please add a description of the reasons in detail for exclusion.
Reply: We have added flow chart (Figure 1. The study flow chart).

3. It is recommended to add PSI, CURB-65, the combination of presepsin level >754 pg/mL and PSI, and the combination of presepsin level >754 pg/mL and PSI for prediction of 30-day mortality in Figure 2.
Reply: We added the AUCs of PSI, CURB65 score, presepsin, and combination of presepsin > 754 and PSI in Figure 3.

4. It is recommended that the methodology used to compare laboratory results between low, moderate, and high risk be described in Table 2.
Reply: The Student’s t test or Mann–Whitney U test was used to compare between survivors and nonsurvivors. The one-way analysis of variance or Kruskal–Wallis test was used for comparison of low-, moderate-, and high-risk groups. We added this information in Section 2.3. Statistical Analysis.

5. In Table 5, the underlying disease was included in the analysis as a risk factor, please specify which underlying conditions were included in the analysis.
Reply: Hypertension, diabetes mellitus, stroke, congestive heart failure, chronic lung disease, chronic liver disease and chronic renal disease were included in the analysis. We added this information in Section 3.4.

6. In Table 4 and Table 5, the cut-off value of PSI is different, why?
Reply: In Table 4 and 5, cutoff value of PSI was corrected to > 126. In addition, the cutoff value of CURB65 score was also corrected to > 2.
